# Circulating Renalase as Predictor of Renal and Cardiovascular Outcomes in Pre-Dialysis CKD Patients: A 5-Year Prospective Cohort Study

**DOI:** 10.3390/life11030210

**Published:** 2021-03-08

**Authors:** Ana Cerqueira, Janete Quelhas-Santos, Inês Ferreira, Susana Sampaio, Miguel Relvas, Nídia Marques, Cláudia Camila Dias, Manuel Pestana

**Affiliations:** 1Nephrology Department, Centro Hospitalar Universitário São João, 4200-319 Porto, Portugal; inescastroferreira@sapo.pt (I.F.); susana.sampaio@sapo.pt (S.S.); miguel.carvalho@chsj.min-saude.pt (M.R.); nidia.marques@chsj.min-saude.pt (N.M.); mvasconcelos@chsj.min-saude.pt (M.P.); 2Department of Medicine, Faculty of Medicine, University of Porto, 4200-250 Porto, Portugal; sjanete@med.up.pt; 3Institute for Innovation and Health Research (I3S), Institute of Biomedical Engineering (INEB), Nephrology and Infectious Diseases Research Group, University of Porto, 4200-250 Porto, Portugal; 4Department of Community Medicine Health Information and Decision, Faculty of Medicine, University of Porto, 4200-250 Porto, Portugal; camila@med.up.pt; 5CINTESIS–Center for Health Technology and Services Research, 4200-250 Porto, Portugal

**Keywords:** chronic kidney disease, CKD progression, cardiovascular risk, renalase, MACCEs

## Abstract

Chronic kidney disease (CKD) is an independent risk factor for adverse cardiovascular and cerebrovascular events (MACCEs), and mortality since the earlier stages. Therefore, it is critical to identify the link between CKD and cardiovascular risk (CVR) through early and reliable biomarkers. Acknowledging that CKD and CKD progression are associated with increased sympathetic tone, which is implicated in CVR, and that renalase metabolizes catecholamines, we aimed to evaluate the relationship between renalase serum levels (RNLS) and cardiovascular and renal outcomes. The study included 40 pre-dialysis CKD patients (19F:21M) with median age of 61 (IQ 45–66) years. At baseline, we measured RNLS as well as routine biomarkers of renal and cardiovascular risk. A prospective analysis was performed to determine whether RNLS are associated with CKD progression, MACCEs, hospitalizations and all-cause mortality. At baseline, the median level of RNLS and median estimated glomerular filtration rate (eGFR) were 63.5 (IQ 48.4–82.7) µg/mL and 47 (IQ 13–119) mL/min/1.73 m^2^, respectively. In univariate analysis, RNLS were strongly associated with eGFR, age and Charlson Index. Over the course of a mean follow-up of 65 (47 to 70) months, 3 (7.5%) deaths, 2 (5%) fatal MACCEs, 17 (42.5%) hospital admissions occurred, and 16 (40%) patients experienced CKD progression. In univariate analysis, RNLS were associated with CKD progression (*p* = 0.001), hospitalizations (*p* = 0.001) and all-cause mortality (*p* = 0.022) but not with MACCEs (*p* = 0.094). In adjusted analysis, RNLS predicted CKD progression and hospitalizations regardless of age, Charlson comorbidity index, cardiovascular disease, hypertension, diabetes and dyslipidemia. Our results suggest that RNLS, closely related with renal function, might have a potential role as predictor of renal outcomes, hospitalizations, and mortality in pre-dialysis CKD patients.

## 1. Introduction

Patients with chronic kidney disease (CKD) present markedly increased rates of mortality and cardiovascular disease (CVD) and most of these patients are more likely to die than progress to end-stage renal disease (ESRD) [1,2]. In addition, CKD progression is also associated with increased CVD risk and CV mortality [3], even after adjusting for the coexisting multiple traditional and nontraditional risk factors [4,5,6,7,8,9,10,11]. Enhanced sympathetic activity is observed in CKD and it has been clearly shown that it is an important predictor of both mortality and increased risk of CVD [12]. In 2005, during the search to identify proteins secreted by the kidney, that could help explain the high incidence of CVD in patients with CKD and represent new therapeutic targets; a team of researchers by Yale University discovered renalase (RNLS). Renalase is secreted by the kidneys into both the circulatory stream and urine, where it could metabolize catecholamines [12,13]. Levels of RNLS have been reported to be higher in patients with hypertension than healthy individuals [14] and positively associated with blood pressure in clinical studies [15]. In addition, recent studies including non-CKD patients point to a correlation between RNLS and atrial fibrillation, advanced left auricular remodeling [16], coronary artery disease, decreased ejection fraction [14] and to an increased risk of myocardial infarction and stroke [17]. One retrospective study including CKD patients demonstrated an association between RNLS and all-cause mortality and adverse renal outcomes [18].

In the present study, we aimed to analyze, in a cohort study, the association of circulating RNLS with renal function and other biomarkers of CV risk, in pre-dialysis CKD patients and prospectively analyze the role of RNLS as a predictor of cardiovascular and renal risk, assessing its relationship with hard outcomes such as progression of CKD, hospitalizations, major adverse cardiovascular/cerebrovascular events (MACCEs) and mortality.

## 2. Methods

The study population included a cohort of 40 pre-dialysis CKD patients followed-up in the outpatient clinic of the Nephrology Department of São João University Hospital Centre. Patients with acute kidney injury, recent hospital admission (<2 weeks), active or recent infections (<1 week) and known psychiatric disturbances were excluded from the study.

At baseline, the etiology of the renal disease, CKD stage classification and a validated comorbidity index (Charlson Index) were determined in all recruited patients. Anthropometric measurements, systolic and diastolic blood pressure (mean of 3 measurements) and serum levels of creatinine, urea, phosphate, parathormone, C reactive protein and proteinuria were evaluated. In addition, blood samples were collected for assessment of RNLS using Uscn Life Science Inc. (Wuhan, China) ELISA kit.

All selected patients were prospectively followed up for a median of 65 (47–70) months, to evaluate hard renal and CV outcomes including progression of CKD and ESRD, hospitalizations, MACCEs as well as CV and all-cause mortality. Data on the occurrence of death, MACCEs [acute coronary syndrome (ACS), heart failure and stroke], hospitalizations and CKD progression were obtained through regular follow-up, complemented by electronic medical records. Renal outcomes included CKD progression, defined as serum creatinine doubling or a >50% decrease in eGFR according to CKD-EPI formula, and renal replacement therapy initiation (ESRD) after enrolment.

### Statistical Analysis

Continuous variables were described as minimum, percentile 25, median, percentile 75 and maximum deviations, and categorical variables were presented as absolute (*n*) and relative frequencies (%). Differences in continuous variables were assessed by Mann–Whitney *U* test, while Chi-square tests were used to analyze differences in categorical variables. A correlation analysis was performed using Spearman correlation coefficients. Logistic regression models were used to visualize the relationships between RNLS and CKD progression and hospitalizations. A *p* < 0.05 was considered statistically significant.

## 3. Ethics

The research was approved by the Ethics Committee for Health and the Local Institutional Review Board of São João University Hospital Centre (CES 251.14) and was carried out in accordance with the Declaration of Helsinki (2008) of the World Medical Association.

## 4. Results

A total of 40 pre-dialysis CKD patients (median age of 61 years; IQ 45–66), male/female (21/19), were enrolled in the study. The baseline demographic, clinical and analytical characteristics of the recruited patients are summarized in Table 1.

The median eGFR was 47 (13–119) mL/min/1.73 m^2^: seventeen patients were included in stages 1–2, nine patients in stages 3a–3b and fourteen patients in stages 4–5. The most frequent etiology of CKD was diabetic nephropathy (15%). Body mass index (BMI) was 27.5 (IQ 25–30) kg/m^2^ and median Charlson index was 4.5 (IQ 2.0–6.0).

Mean RNLS were 65.5 (IQ 48.2–82.69) µg/mL. Renalase levels were significantly higher in patients with more advanced CKD stages, and were closely related with eGFR decline (Figure 1). In addition, RNLS were negatively correlated with hemoglobin levels (r = −0.360, *p* = 0.023) and HDL cholesterol (r = −0.455, *p* = 0.004) and were positively correlated with age (r = 0.407, *p* = 0.009), Charlson Index (r = 0.704, *p* < 0.001), serum urea (r = 0.818, *p* < 0.001), serum creatinine (r = 0.877, *p* < 0.001), serum phosphate (r = 0.590, *p* < 0.001), iPTH (r = 0.694, *p* < 0.001), triglycerides (0.383, *p* = 0.016), uric acid (r = 0.565, *p* < 0.001) and BNP (r = 0.546. *p* = 0.003) (Table 2). Baseline cardiovascular risk factors, including hypertension, diabetes, dyslipidemia or cerebrovascular disease did not significantly associate with RNLS. However, patients with cardiovascular disease had significantly higher RNLS (59.44 vs. 76.64 µg/mL, *p* = 0.028) (Table 3).

Over the course of a median follow-up period of 65 (47 to 70) months, 3 patients died, 2 patients suffered a fatal MACCE, 17 patients had at least one hospital admission and 16 patients experienced CKD progression, of which 12 progressed to ESRD and started renal replacement therapy (RRT) (Table 4). Renalase levels were higher at baseline in patients with CKD progression during follow-up (median 80.40 vs. 51.79, *p* = 0.001), in patients with hospitalizations (median: 81.43 vs. 53.86, *p* = 0.001) and in patients who died (median: 95.20 vs. 60.22, *p* = 0.022) (Table 5). In adjusted analysis, RNLS predicted CKD progression and hospitalizations regardless of age, Charlson comorbidity index, cardiovascular disease, hypertension, diabetes and dyslipidemia (Table 6). The small number of events (*n* = 3) prevented us from performing an adjusted analysis to mortality.

## 5. Discussion

In the present study, we investigated the association between RNLS and traditional CV risk factors including eGFR in a cohort of non-dialysis CKD patients. In addition, we examined the role of RNLS as a biomarker of CV and renal outcomes in this pre-dialysis CKD patient population, prospectively followed-up for more than 5 years. Our main findings were the following: (i) RNLS are closely correlated with the decrease in eGFR in pre-dialysis CKD patients; (ii) RNLS are associated with cardiovascular disease in pre-dialysis CKD patients; (iii) Pre-dialysis CKD patients with higher RNLS at baseline presented higher rates of progression of CKD, hospitalizations and mortality, but not MACCEs.

In the present study, we found that RNLS assessed by a commercially available ELISA kit were strongly and inversely associated with eGFR. Our results agree well with the previous observations from our and other groups that reported that RNLS assessed by ELISA assay with specific monoclonal antibody are negatively associated with renal function. In the recent literature, seven studies were carried out in CKD patients stages 1 to 4, and five of them provided evidence for a negative relationship between RNLS and eGFR (Table 7). The other two studies reporting discrepant results used different commercial ELISA kits to access RNLS, which may justify the different observations (Table 7). Findings similar to ours showing inverse correlation between RNLS and eGFR were also reported in other CKD populations, including kidney transplant recipients [19,20], patients on peritoneal dialysis [21] and patients on hemodialysis [22] (Table 7).

Because our study included patients with stages 1 to 5 CKD, the strong and inverse association observed between circulating RNLS and renal function reinforces the view that RNLS may represent a useful biomarker for early identification of renal dysfunction in stable pre-dialysis CKD patients.

In our study baseline, CVR factors including hypertension, diabetes, dyslipidemia or cerebrovascular disease did not significantly associate with higher RNLS. There is controversy in the literature regarding the association of hypertension and RNLS. The lack of association between hypertension and RNLS in our study can be explained on the basis that most patients were considered hypertensive because they were on anti-hypertensive medication, although their average blood pressure levels were near normal.

In our pre-dialysis CKD population, patients with cardiovascular disease had significantly higher RNLS. This agrees well with the findings by Gluba-Brzózka et al., that reported higher RNLS in CKD patients with coronary artery disease and further suggested a possible role of RNLS in its pathogenesis.

Our patient population was prospectively followed-up for more than 5 years. During this period, we found that 42.5% of the patients had at least one hospital admission for medical cause, 40% suffered a significant decline in eGFR and 30% progressed to ESRD. Nevertheless, only 3 deaths where registered, of which 2 were from MACCEs. In univariate analysis, positive associations were observed between baseline RNLS and all-cause mortality, hospitalizations, and CKD progression but not with the occurrence of MACCEs. These results are in line with those observed in the retrospective analysis of the data from the K-STAR study including 383 pre-dialysis CKD patients. The authors reported that RNLS were not associated with the occurrence of MACCEsl; nevertheless, they found that each 10 μg/mL increase in RNLS was associated with a significantly greater hazard of all-cause mortality and adverse renal outcomes [18]. In adjusted analysis, RNLS predicted CKD progression and hospitalizations regardless of age, Charlson index, cardiovascular disease, diabetes, hypertension or dyslipidemia. It should be emphasized that our results were obtained in a cohort of pre-dialysis CKD patients that were already on renoprotective therapy (95% were on ACEI/ARBs). Taken together, our findings suggest that RNLS can be, not only a biomarker of renal function but, may also provide valuable information in prediction of relevant outcomes in CKD patient population.

We acknowledge some limitations of our study. First, this is a single-center study with a relatively small number of patients and the inherent selection bias; second, we did not enroll healthy subjects as controls for comparing RNLS to those with established CKD; third, the small number of outcomes observed, notwithstanding the 5 years of follow-up, may have jeopardized the analysis. Nonetheless, to our knowledge, this is the first study to prospectively assess the role of RNLS as predictor of CV and renal outcomes in a pre-dialysis CKD population.

In conclusion, our results show that RNLS are closely related with renal function in pre-dialysis CKD patients. It is also suggested that the role of circulating RNLS may be a useful biomarker to predict renal outcomes, hospitalizations and mortality in pre-dialysis CKD patients. Further studies are needed to explore the possible role of RNLS as a new therapeutic target in the prevention and treatment of CKD and cardiovascular diseases.

## Figures and Tables

**Figure 1 life-11-00210-f001:**
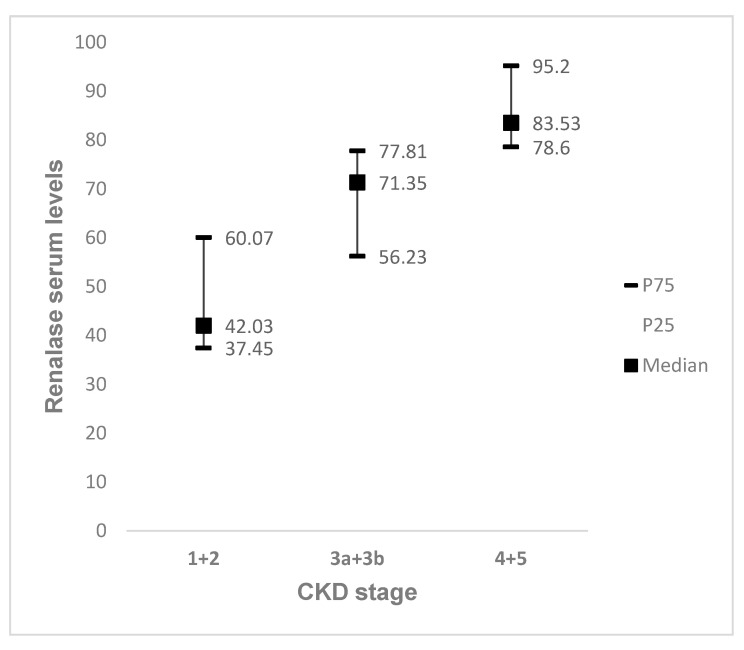
Circulating renalase levels according to increased CKD stages. Significant differences between CKD 4 + 5 and CKD 1 + 2 (*p* < 0.05).

**Table 1 life-11-00210-t001:** Baseline demographic, clinical and analytical characteristics of the study population (*n* = 40).

Variable		
Follow-Up (months), median (P25–P75)	65.4	(57.4–68.6)
Demographic data		
Gender (male), *n* (%)	21	(52.5)
Age (years), median (P25–P75)	61	(45–66)
Charlson Index, median (P25–P75)	4.5	(2–6)
Clinical data		
Systolic blood pressure (mmHg), (P25–P75)	137	(117–150)
Diastolic blood pressure (mmHg), (P25–P75)	76	(66–83)
Body mass index (kg/m^2^), median (P25–P75)	27.5	(25–30)
ACEI/ARBs, *n* (%)	38	(95.0)
Renalase levels (ug/mL), median (P25–P75)	63.53	(48.42–82.69)
CKD related parameters		
Diabetic Nephropathy n (%)	6	(15.0)
eGFR CKD–EPI (mL/min/1.3 m^2^), median (min–max)	47	(13–119)
Creatinine (mg/dL), median (P25–P75)	1.44	(1.02–2.46)
Urea (mg/dL), median (P25–P75)	73.5	(44.5–109.0)
Calcium (mg/dL), median (P25–P75)	4.75	(4.60–4.90)
Phosphate (mg/dL), median (P25–P75)	3.25	(2.90–4.00)
Parathormone (pg/mL), median (P25–P75)	97.0	(45.0–120.0)
25-OH-Vitamin D (ng/mL), median (P25–P75)	20	(11–31)
Protein/creatinine ratio (mg/g), median (P25–P75)	455.5	(154.2–966.8)
Cardiovascular related parameters		
Hemoglobin (g/dL), median (P25–P75)	12.7	(11.60–13.75)
Albumin (g/dL), median (P25–P75)	41.05	(39.20–43.55)
Total Cholesterol (mg/dL), median (P25–P75)	184	(155–207)
HDL Cholesterol (mg/dL), median (P25–P75)	49	(41–59)
Triglycerides (mg/dL), median (P25–P75)	121	(95–180)
Uric Acid (mg/dL), median (P25–P75)	6.4	(5.3–8.7)
C reactive protein (mg/L), median (P25–P75)	2.2	(1.2–5.7)
BNP (pg/mL), median (P25–P75)	66.0	(24.0–177.0)

eGFR: estimated glomerular filtration rate; BNP: B-type natriuretic peptide; ACEI—Angiotensin Converting Enzyme Inhibitors ARBs—Angiotensin II receptor blockers.

**Table 2 life-11-00210-t002:** Associations between RNLS and Routine Biomarkers of Renal and Cardiovascular Risk.

	Renalase	
	Spearman Correlation	*p*-Value
Demographic data		
Age (years)	0.407	**0.009**
Charlson Index	0.704	**<0.001**
CKD related parameters		
eGFR CKD-EPI (mL/min/1.73 m^2^)	−0.883	**<0.001**
Creatinine (mg/dL)	0.877	**<0.001**
Urea (mg/dL)	0.818	**<0.001**
Protein/creatinine ratio (mg/g)	0.133	0.426
Phosphate (mg/dL)	0.590	**<0.001**
Parathormone (pg/mL)	0.694	**<0.001**
25-OH-Vitamin D (ng/mL)	0.163	0.357
Cardiovascular related parameters		
Hemoglobin (g/dL)	−0.360	**0.023**
Albumin (g/dL)	−0.095	0.559
Total Cholesterol (mg/dL)	−0.236	0.480
HDL Cholesterol (mg/dL)	−0.455	**0.004**
Triglycerides (mg/dL)	0.383	**0.016**
Uric Acid (mg/dL)	0.565	**<0.001**
C reactive protein (mg/L)	0.153	0.372
BNP (pg/mL)	0.546	**0.003**

eGFR: estimated glomerular filtration rate; BNP: B-type natriuretic peptide.

**Table 3 life-11-00210-t003:** Associations between RNLS and patients’ comorbidities.

	Renalase				
	P25	Median	P75	*n*	*p*-Value
**Hypertension**					
No	33.94	45.51	83.88	4	0.279
Yes	49.18	68.17	82.69	36	
**Diabetes**					
No	42.03	61.42	78.60	29	0.209
Yes	54.06	65.63	99.54	11	
**Dyslipidemia**					
No	41.20	51.69	77.62	16	0.104
Yes	56.23	71.35	83.74	23	
**Cardiovascular disease**					
No	40.95	59.44	78.60	29	**0.028**
Yes	70.71	76.64	86.61	11	
**Cerebrovascular disease**					
No	47.98	60.82	83.31	38	0.420
Yes	75.64	78.54	81.43	2	

**Table 4 life-11-00210-t004:** Cardiovascular and renal outcomes during follow-up in the studied population.

Outcome	*n* (%)
MACCEs	2 (5.0)
Acute myocardial infarction	1 (2.5)
Stroke	1 (2.5)
Hospital admission for medical causes	17 (42.5)
Death	3 (7.5)
Death by MACCEs	2 (5)
CKD progression	16 (40.0)
RRT	12 (30.0)

MACCEs: major adverse cardiovascular and cerebrovascular events; RRT: Renal Replacement Therapy.

**Table 5 life-11-00210-t005:** Association between RNLS and outcomes.

	Renalase	
	Median (P25–P75)	*p*-Value ^1^
CKD progression		**0.001**
No	51.79 (40.6–73.5)	
Yes	80.4 (62.93–88.07)	
Mortality		**0.022**
No	60.22 (48–78.6)	
Yes	95.20 (81.4–113.9)	
Hospitalization		**0.001**
No	53.86 (40.4–71.4)	
Yes	81.43(65.6–95.1)	
MACCEs		0.094
No	60.82 (48–82.1)	
Yes	95.12 (81.4–108.1)	

1—Mann–Whitney test; MACCES: major adverse cardiovascular and cerebrovascular events.

**Table 6 life-11-00210-t006:** Linear regression: association between CKD progression and hospitalizations (dependent variables) and renalase, adjusted to renal function, age, Charlson comorbidity index, cardiovascular disease, hypertension, diabetes and dyslipidemia.

	CKD Progression			Hospitalizations		
	OR	IC 95%	*p*	OR	IC 95%	*p*
**Model 1**						
Renalase	1.055	1.015–1.096	0.007	1.071	1.025–1.119	**0.002**
Hosmer Lemeshow *p*-value	0.565			0.797		
**Model 2**						
Renalase	1.064	1.019–1.112	0.005	1.074	1.025–1.126	**0.003**
Age	0.971	0.921–1.023	0.265	0.988	0.937–1.041	0.645
Hosmer Lemeshow *p*–value		0.095			0.673	
**Model 3**						
Renalase	1.050	1.001–1.101	0.044	1.062	1.008–1.119	**0.023**
Charlson comorbidity index	1.060	0.715–1.570	0.773	1.117	0.740–1.684	0.599
Hosmer Lemeshow *p*-value		0.840			0.521	
**Model 4**						
Renalase	1.068	1.021–1.117	0.004	1.067	1.020–1.116	**0.005**
Cardiovascular disease	0.269	0.045–1.607	0.150	1.518	0.287–8.028	0.623
Hosmer Lemeshow *p*-value	0.646			0.848		
**Model 5**						
Renalase	1.054	1.015–1.095	0.007	1.070	1.024–1.117	**0.002**
Hypertension	2.261	0.084–60.550	0.627	2.954	0.057–153.369	0.591
Hosmer Lemeshow *p*-value	0.477			0.789		
**Model 6**						
Renalase	1.055	1.012–1.101	0.012	1.074	1.023–1.127	**0.004**
Diabetes	3.333	0.635–17.502	0.155	3.050	0.517–17.984	0.218
Hosmer Lemeshow *p*-value	0.775			0.775		
**Model 7**						
Renalase	1.052	1.012–1.095	0.011	1.072	1.026–1.121	**0.002**
Dyslipidemia	2.413	0.510–11.423	0.267	0.797	0.157–4.058	0.785
Hosmer Lemeshow *p*-value	0.561			0.461		

**Table 7 life-11-00210-t007:** Data from the most Relevant and Recent Studies Evaluating Renalase by ELISA in CKD Patients.

	Levels of Circulating Renalase by Elisa Kit	Correlation with Renal and CV Outcomes
**CKD Patients Stages 1–4**		
A. Gluba-Brzózka et al. 2014 [23] 139 CKD patients 45 healthy volunteers	Increased concentrations of renalase control vs. CKD group 251.0 ± 157 vs. 316.1 ± 155.3 ng/mL, *p* = 0.026 *USCN Life Science, E92845Hu*	Increased concentration of osteocalcin, renalase, MMP-2 and TIMP-2 suggest that these factors may be involved in the pathogenesis of CAD in patients with CKD.
J. Quelhas-Santos et al. 2014 [20] 26 ESRD patients	Plasma renalase levels (ug/mL) 4.7 ± 0.5 LKD 29.4 ± 4.0 LKR before TX (*p* < 0.05) *USCN Life Science, E92845Hu*	Plasma renalase levels closely depend on renal function and sympathetic nervous system activity.
F. Wang et al. 2015 [15] 87 CKD patients stages I to IV	Renalase levels not different between groups CKD1–2 (162.1 ± 40.1 ng/L) vs. healthy control group (167.8 ± 69.4 ng/L) group CKD3–5 (217.4 ± 103.8 ng/L) were significantly increased compared with group CKD1–2 (*p* < 0.05) *ELISA Yaji Biological Corp*	Serum renalase levels were positively correlated with CKD stage (*p* < 0.05), while negatively correlated with eGFR (*p* < 0.05)
S. H. Baek et al. 2019 [18] 383 patients with CKD	Mean level of serum renalase was 75.8 ± 34.8 μg/mL *ELISA Cloud Clone Corp*	Higher serum creatinine levels were significantly associated with a higher renalase levels. Increase in serum renalase was associated with all-cause mortality and adverse renal outcomes, but not associated with the rate of MACCEs.
P. Skrzypczyk et al. 2019 [24] 38 children with CKD (stage G2-5) 38 healthy children	Renalase level was higher in the study group compared to control group values (*p <* 0.001) *ELISA CloudClone Corp*	In multivariate analysis GFR (*β* = −0.63, *p <* 0.001), was determinant of renalase In children with CKD there is a strong correlation between renalase level and CKD stage.
M. Wiśniewska et al. 2019 [25] 155 white patients with CKD 30 healthy controls	Serum renalase levels were higher in patients with CKD than in controls: median (Q1-Q3), 103 ng/mL (55.6–166 ng/mL) vs. 17.7 ng/mL (16.3–21.8 ng/mL); *p* < 0.001 *ELISA kit EIAab*	No association between serum renalase and eGFR. No associations were found between renalase concentrations and other causes of CKD.
N. M. Serwin et al. 2020 [26] 62 CKD patients stages I to IV 28 healthy controls	The concentration of renalase in the serum of CKD patients was much higher in comparison to material from healthy individuals 36.1 (18.3–109.1) vs. 11.1 (2.5–26.5) ng/mL *ELISA kit EIAab*	Renalase levels in serum are not related to the glomerular filtration rate.
**HD and DP Patients**		
E. Zbroch et al. 2012 [22] 104 HD patients	Mean serum renalase in the study cohort was significantly higher than in the control group (27.53 ± 7.18 vs. 3.86 ± 0.73 µg/mL, *p* < 0.001) *USCN Life Science, E92845Hu*	Significant inverse correlation between the serum renalase and residual renal function (r = −0.327, *p* = 0.001). Renalase was not related to blood pressure, heart rate or hemodialysis vintage.
J. Malyszko et al. 2012 [27] 34 HD patients	Mean serum renalase concentration in the study cohort was 17.51 6.73 μg/mL and it was significantly higher when compared with the healthy volunteers—3.99 1.73 μg/mL (*p* < 0.001) *USCN Life Science, E92845Hu*	Serum renalase correlated with creatinine (r = 0.43, *p* < 0.05), residual renal function (r = 0.39, *p* < 0.05). The only predictor of renalase in multiple regression analysis was the presence of hypertension explaining 90% of the renalase variations.
E.-Zorawska et al. 2012 [28] 60 HD patients	Mean level of renalase was significantly higher in HD patients when compared to the control group (27.53 ± 9.39 µg/mL vs. 4.00 ± 1.37 µg/mL, *p* < 0.001 *USCN Life Science, E92845Hu*	Renalase appeared to be unrelated to Vascular adhesion protein-1.
E. Zbroch et al. 2012 [21] 26 PD patients	Serum concentration of renalase was significantly higher in patients dialyzed for more than 6 months than in those dialyzed for fewer than 6 months (21.15 ± 4.58 μg/mL vs. 16.63 ± 2.86 μg/mL, *p* = 0.008) *USCN Life Science, E92845Hu*	Renalase was not related to BP control, BP level, sex, dialysis adequacy, or residual renal function.
E. Zbroch et al. 2013 [29] 75 HD patients 26 PD patients	HD patients had higher renalase levels (27.49 ± 6.9 ug/mL) Renalase were higher in dialyzed groups (19.24 ± 4.5 ug/mL) comparing to healthy volunteers (3.86 ± 0.74 ug/mL) *USCN Life Science, E92845Hu*	Renalase correlated with dialysis vintage and inversely with residual diuresis. HD population with CAD had higher renalase level than their PD counterparts.
M. Dziedzic et al. 2014 [30] 49 HD patients	The mean concentration of renalase in the entire study population was 126.59 ± 32.63 ng/mL *USCN Life Science, E92845Hu*	Inverse correlation between NT-proBNP and renalase plasma levels in HD patients were due to impaired kidney function, accompanied by increased sympathetic nerve activity, which have an impact on the development of hypertension and cardiovascular complications.
E. G. Oguz et al. 2016 [31] 50 HD patients 35 healthy controls	Serum renalase levels were significantly higher in HD patients (212 ± 127 ng/mL) compared to controls (116 ± 67 ng/mL) (*p* < 0.001). *USCN Life Science, E92845Hu*	Renalase was positively correlated with serum creatinine and dialysis vintage (r = 0.677, *p* <0.001 and r = 0.625, *p* < 0.001, respectively). There was no significant association of renalase with LVMI in the HD patients (r = 0.263, *p* = 0.065).
E. G. Oguz et al. 2017 [32] 40 PD patients 40 healthy controls	Serum renalase level was significantly higher in the PD patients than in the control group [176.5 (100–278.3) vs. 122 (53.3–170.0) ng/mL] (*p* = 0.001) *USCN Life Science, E92845Hu*	Renalase was negatively correlated with RRF (r = −0.511, *p* = 0.021). Renalase is associated with residual renal function but not with CVD risk factors in PD patients.
M. Wisniewska et al. 2021 [33] 77 HD patients 30 healthy controls	Renalase serum concentrations in CKD patients were significantly increased when compared with control subjects (185.5 ± 64.3 vs. 19.6 ± 5.0 ng/mL; *p* < 0.00001 *ELISA kit EIAab*	The decreased plasma concentrations of catecholamines may be due to their increased degradation by plasma renalase.
**Renal Transplant**		
J. Malyszko et al. 2011 [19] 89 kidney allograft recipients 27 healthy volunteers	The mean serum renalase among recipients was significantly higher compared with the control group (6.72 ± 4.50 µg/mL vs. 3.86 ± 0.73 µg/mL; *p* < 0.001) *USCN Life Science, E92845Hu*	In kidney transplant recipients, renalase correlated serum creatinine (r = 0.49; *p* < 0.001) and estimated glomerular filtration rate r = −0.44; *p* < 0.0001
E. Zbroch et al. 2012 [34] 62 kidney allograft recipients 27 healthy volunteers	The mean serum renalase level in kidney allograft recipients was significantly higher compared with the control group (6.72 ± 2.86 µg/mL vs. 3.86 ± 0.73 µg/mL, *p* < 0.001 *USCN Life Science, E92845Hu*	In hypertensive allograft recipients, renalase was significantly higher than in normotensives. A multiple regression analysis showed that renalase was predicted in 58% by serum creatinine.
D. Stojanovic et al. 2015 [35] 73 renal TX recipientes	Renalase ng/mL Renal transplant recipients (141.82 ± 36.47) Control group (16.36 ± 4.13) *USCN Life Science, E92845Hu*	Significant risk of reduced glomerular filtration rate in transplant recipients with increased renalase concentration (*p* = 0.026). Renalase was shown to be strong predictor of decreased glomerular filtration rate.
D. Stojanovic et al. 2017 [36] 73 renal TX recipients	Plasma renalase level was increased compared to controls, 141.82 ng/mL vs. 16.36 ng/mL, *p* < 0.0001 *USCN Life Science, E92845Hu*	Significant inverse correlation between renalase and estimated glomerular filtration rate (r = −0.552, *p* < 0.001)

## Data Availability

The data presented in this study are available on request from the corresponding author.

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
