# Peer review of "Circulating Renalase as Predictor of Renal and Cardiovascular Outcomes in Pre-Dialysis CKD Patients: A 5-Year Prospective Cohort Study"

_life, 2021, doi:10.3390/life11030210_

Round 1

Reviewer 1 Report

The topic of the work is very interesting, especially for doctors treating patients with chronic kidney disease. However, the obtained results of the analyzes and their discussion are quite general. I suggest more perspicacious discussion obtained results.

Reviewer 2 Report

Thank you for opportunity for reviewing this paper “Circulating Renalase as Predictor of Renal and Cardiovascular Outcomes in Pre-Dyalisis CKD Patients: A 5 Year Prospective Study”. This article is interesting, but some issues should be explained more.

1.The authors should explain why using cross-sectional study design instead of cohort study due to this long-term follow-up period.

2.Please give more information that how do you calculate the sample sizes?

3.A statement including the reference number of the ethics committee where appropriate should appear in the manuscript.

4.Please consider the comparison with the other studies in other areas using table so make clear the significance of this study.

5. How physicians or policy makers can deliberate with patients or people based on the key findings of this paper?

6.The goodness-of-fit test is also suggested in the logistic regression.

7.The authors should add the comments related to selection bias in this study to the perceived limitation subsection.

8.Some references should be updated. In addition, please make sure whether formats are described according to the instructions for authors.

If the above suggestions are incorporated and the paper is thoroughly edited, it will be a strong contribution to the literature.
